# Microsecond All-Optical Modulation by Biofunctionalized Porous Silicon Microcavity

**DOI:** 10.3390/nano13142070

**Published:** 2023-07-14

**Authors:** Dániel Petrovszki, Sándor Valkai, Lóránd Kelemen, László Nagy, Vivechana Agarwal, Szilvia Krekic, László Zimányi, András Dér

**Affiliations:** 1Institute of Biophysics, Biological Research Centre, Eötvös Loránd Research Network, 6726 Szeged, Hungary; danielpetrovszki37@gmail.com (D.P.); valkai.sandor@brc.hu (S.V.); kelemen.lorand@brc.hu (L.K.); krekic.szilvia@brc.hu (S.K.); 2Doctoral School of Multidisciplinary Medical Science, University of Szeged, 6720 Szeged, Hungary; 3Department of Medical Physics and Informatics, Faculty of Science and Informatics, Albert Szent-Györgyi Medical School, University of Szeged, 6720 Szeged, Hungary; l.nagy@physx.u-szeged.hu; 4Institute of Plant Biology, Biological Research Centre, Eötvös Loránd Research Network, 6726 Szeged, Hungary; 5Centro de Investigación en Ingeniería y Ciencias Aplicadas-IICBA, Universidad Autónoma del Estado de Morelos, Cuernavaca 62209, Morelos, Mexico; vagarwal@uaem.mx

**Keywords:** porous silicon, photoactive yellow protein, photocycle, optical modulation

## Abstract

We successfully created a composite photonic structure out of porous silicon (PSi) microcavities doped by the photochromic protein, photoactive yellow protein (PYP). Massive incorporation of the protein molecules into the pores was substantiated by a 30 nm shift of the resonance dip upon functionalization, and light-induced reflectance changes of the device due to the protein photocycle were recorded. Model calculations for the photonic properties of the device were consistent with earlier results on the nonlinear optical properties of the protein, whose degree of incorporation into the PSi structure was also estimated. The successful proof-of-concept results are discussed in light of possible practical applications in the future.

## 1. Introduction

### 1.1. Optical Switching by Chromoproteins

The nonlinear optical (NLO) properties of dried chromoprotein samples have recently been intensively investigated for their possible applications as NLO materials in integrated optical (IO) devices [1,2,3,4,5,6,7,8]. Most of the potential applications have been envisioned using bacteriorhodopsin- (bR-) based films, as they undergo high-refractive index changes (up to 5 × 10^−3^) upon photoexcitation [1,2,3,9], at the same time, showing high cyclicity (>10^6^) and considerable long-term stability [10,11,12].

More recently, dried samples from other chromoproteins, most notably photoactive yellow protein (PYP), have also been proven to possess NLO properties making them suitable for use as NLO materials in future integrated optic applications [13,14,15,16]. PYP is a 14 kDa water-soluble protein that drives negative phototaxis in purple sulfur bacteria, such as *Halorhodospira halophila* [17]. PYP is a prominent member of the so-called Per–Arnt–Sim (PAS)-domain protein superfamily. PAS domains are characteristic primary motifs of signaling pathways of sensory proteins that can be found in all kingdoms of life. Besides their sensory function, they mediate protein–protein interactions as well. PYP displays the same α/β fold structure, with a central β-sheet with five strands and helical connectors on both sides. The β-sheet separates two hydrophobic cores, one with the chromophore-binding pocket, while the other containing the N-terminus [18]. The chromophore responsible for light absorption is a p-coumaric acid derivative of trans configuration in the ground state [17,18]. Upon light excitation, the chromophore undergoes trans-cis isomerization, triggering a sequence of cyclic protein-conformational changes, called the photocycle. The quasi-stable intermediate states of the photocycle possess distinct spectral properties, which is the main factor when considering PYP for IO applications. PYP films at high enough relative humidity (>60%) preserve the basic scheme of their native photocycle (Figure 1), a series of first-order reactions of metastable conformational states, including intermediates of red- (pR) and blue-shifted (pB) absorption spectra, as compared to that of the resting state (pG).

A row of recent studies present thorough analyses of the unique spectrokinetic properties of PYP films, including their high second-order nonlinearities, responsible for a Kerr-constant (4 × 10^−4^) that is several orders of magnitude higher than conventional nonlinear-optical signals, and a light-induced refractive index change reaching the 10^−3^ magnitude [13,14,15,16]. Based on these observations, PYP films can be considered promising alternatives to bR-based NLO elements in future integrated optical applications. A special advantage of PYP compared to bR resides is its relatively small dimension, water-solubility and, hence, the potential of possible incorporation into porous matrices compatible with integrated optics.

### 1.2. Porous Silicon Structures

Silicon is considered to be one of the main optical materials in contemporary photonics technology, and more specifically in integrated optics. Being transparent at the fiber optics telecom wavelength regimes (centered around 1350 nm and 1550 nm, respectively), and possessing a high-refractive index in this range (n > 3), it is ideally suited for the creation of waveguides and other passive IO elements. Electronic microchip technology has elaborated sophistical tools and methods to create silicon structures of sub-100 nm spatial resolution, which is perfectly suitable for the usually sub-micrometer-sized integrated photonic structures. On the other hand, the existing silicon microfabrication technology allows the creation of hybrid devices by combining integrated photonic and electronic structures [19].

Ideally, the active elements of IO circuits, that play an analogous role to switching transistors in electronics, are also to be built around a silicon base. Although silicon shows a non-negligible optical nonlinearity, based on which successful second- and third-harmonic generation experiments have been carried out [19], intense research is going on to improve its NLO properties using, e.g., ion-implantation, etching, or creating nanocomposite materials of silicon base. Creating porous silicon (PSi) structures is an appealing technology to favourably combine all these methods. Thin (10–100 μm thick) PSi layers can be formed, e.g., using wet electrochemical etching of p-type (usually boron-doped) silicon wafers of proper crystallographic orientation, usually (1, 0, 0), resulting in a porous silicon layer of quasi-parallel through-holes, running perpendicular to the surface of the wafer [19]. The typical diameter of the holes spans from 10 nm to several 100 nm (from “microporous” to “macroporous” via “mesoporous” PSi layers). If, by a photonic application, the wavelength of the incident light (normally in the infrared range) is considerably higher than the pore size, the PSi behaves as a bulk material, whose refractive index decreases with the increase in porosity. By changing the etching current, the diameter of the holes, and, thus, the porosity of the layer, can be controlled, which lends the PSi-technique a special versatility: fine-tuning the refractive index of the layer, creating special multilayer-structures of photonic band-gap properties (“microcavities”), or combining the PSi structure with other materials, by filling up the holes. The high active surface-to-volume ratio of PSi makes it a promising candidate for sensorics application as well [19].

As for the NLO properties, it has been shown that the nonlinear susceptibilities of high-porosity PSi structures could exceed that of crystalline silicon (n_2_ ≈ 5·10^−14^ cm^2^/W) [19], nevertheless, these values are still orders of magnitude lower than the Kerr-constants of bR or PYP-based films. Hence, the question arises whether doping PSi structures with such chromoproteins can result in a nanocomposite material that could potentially be useful for photonics applications. The highly ordered patterned 3D structures with controllable large exposed surface areas can be modulated during the fabrication process, which offers possible decoration with functional materials. By combining with an enhanced yield of immobilization of biomolecules, a wide number of (bio)-hybrid devices can be constructed with extreme sensitivity, selectivity, and specificity. During recent years, several proteins have been successfully incorporated into PSi microcavities, for potential applications in bioelectronics, such as in biosensorics, biophotonics, low-power electronics, or photovoltaics (e.g., for analytical, environmental, or medical purposes) [20,21,22,23,24,25]. Several organic molecules, enzymes (glucose oxidase), light energy converting (photosynthetic), and heat-shock proteins embedded in the micropores of the PSi structures were shown typically to preserve their basic functionalities, occasionally with enhanced properties for special applications [25].

### 1.3. Aim of the Study

Here, we report on creating a hybrid photonic structure, a PSi microcavity doped with the soluble chromoprotein PYP, characterizing its static and light-induced dynamic spectral properties, and discussing the main implications of the results for possible biophotonic applications.

## 2. Materials and Methods

### 2.1. PYP Sample Preparation

The PYP solution preparation followed the process as previously described in numerous studies [14,15,16,17,18]; however, the functionalization protocol for the protein-solid interface was adapted to the PSi substrate.

To promote microcavity infiltration with the protein sample, previously fabricated porous silicon substrates with microcavities were exposed to oxygen plasma at 400 mtorr pressure (@29.6 W RF power) for 60 s (PDC-002 Expanded Plasma Cleaner, Harrick Plasma, Ithaca, NY, USA) hydrophilizing their surfaces. Then, the PYP solution was pipetted on the substrates to fill these cavities. The protein films were left drying for 24 h on the substrates. Finally, the extra volumes of the PYP coatings on the slides were washed mildly using deionized water (MilliQ water, Synergy^®^ UV Water Purification System, Merck-Millipore, Burlington, MA, USA), and the PYP functionalized porous silicon microcavity samples became ready-to-use for the measurements.

### 2.2. Porous Silicon Microcavity (PSiMc) Fabrication

PSiMc-s were prepared in dark using a wet electrochemical etching process of highly boron-doped p-type silicon wafers sliced in the (100) crystallographic orientation. Their thickness was in the 500−550 μm range and the specific resistivity was in the 0.002−0.004 Ωcm range. The electrolyte for etching was a mixture of hydrofluoric acid (48 wt%), ethanol (98%), and glycerol (98%) in a volumetric ratio of 3:7:1. Freshly etched samples were washed with ethanol and dried with pentane. Etching was performed with a current density of 80 mA/cm^2^ (H for high porosity, yielding low effective dielectric constant, n_L_) and 40 mA/cm^2^ (L for low porosity, yielding high effective refractive index, n_H_) [26,27]. The anodization times were set to produce microcavity structures (or Fabry–Perot dielectric filters) of alternating quarter-wave layers with d_H_ and d_L_ thickness (n_H_d_H_ = n_L_d_L_ = λ/4 with the wavelength around 645 nm) in the following sequence: [HL] × 5 [HH] [LH] × 5. The top layer was of high porosity, allowing easy incorporation of PYP macromolecules. This is based on previous studies (e.g., [20]) where it was shown that molecular infiltration is facilitated into the higher porosity layer as compared to the lower porosity layer. The two highly porous layers in the middle (marked by bold face characters), function as an optical cavity with an optical mode corresponding to λ/2 optical thickness (λ = 645 nm) [20].

### 2.3. Scanning Electron Microscopy

Scanning electron microscopic (SEM) images were taken of the PSiMc samples to examine the fine structures both from the top and from the cross-section. The SEM images were taken in high vacuum mode of the instrument (JEOL JSM-7100F-LV, JEOL Corporation, Akishima, Tokyo, Japan) without metal coating of the PSi surface. The accelerating voltage was set to 15 kV for better resolution but when imaging the layer’s cross section, 5 kV acceleration often resulted in images of higher contrast.

### 2.4. Experimental Setup

#### 2.4.1. PYP Infiltrated PSiMc Reflectance Measurements

The reflectance spectra of the silicon wafers were measured in a direct way. The collimated light of a halogen bulb (70 W, 12 V) illuminated one input of a specific dedicated 3-bundle lightguide whose output approached the sample surface to ~1 mm distance, perpendicularly. The spot size of illumination was 2 mm in diameter. The reflected light was collected using the same bundle and the corresponding output (reflected) light was deflected by a prism into a commercial fiber optic. The fiber optic was led into a miniature spectrometer (Scanspec UV/Vis spectrometer, Scansci Ltd., Vila Nova de Gaia, Portugal), controlled using a PC. The schematics of the setup are shown in the following figure (Figure 2).

The reflectance spectra of the PSiMc before and after the infiltration by PYP were measured relative to a reference intensity spectrum measured on the Si wafer outside of the porous area. The micrometric X-Y stage allowed to register the reflectance spectrum at several reproducible positions over the porous area before and after PYP infiltration.

From the shifted reflectance spectrum, it could be determined at what wavelength regime the highest modulation can be expected due to the refractive index change upon excitation of the PYP molecules in the porous silicone layers, hence, the wavelength of the probe beam could be selected (see next section).

#### 2.4.2. Photoinduced All-Optical Modulation

The all-optical modulation by the PYP-filled PSiMc-s was based on the photoreaction cycle of PYP (Figure 1). To characterize its kinetics, a time-resolved reflectance measurement setup was designed (Figure 3). In this optical system a Nd:YAG laser (Continuum Surelite II, Amplitude Laser Inc., Milpitas, CA, USA) extended with an optical parametric oscillator (Surelite OPO, Amplitude Laser Inc., Milpitas, CA, USA) was used as a pump source. The probe beam was a continuous red-light (λ = 675 nm) laser directed perpendicularly onto the sample. The pulse energy of the pump was 26 mJ (7 ns pulsewidth, 1.43 Hz), and its wavelength was tuned to 460 nm for exciting the resting state of PYP. The spatial overlapping of the corresponding beams on the sample was achieved by mirrors and an iris diaphragm. The laser power on the sample for the probe was 14.64 mW/cm^2^. The angle difference of the probe beam was ~6.6° relative to the pump. The beam diameters at the beam waist were 2 mm for both beams. The PYP-functionalized porous silicon microcavity sample was placed on a motorized stage with a micropositioner (DC-3 K, Märzhäuser Wetzlar GmbH & Co. KG, Wetzlar, Germany) to obtain precise and reproducible illumination location over the porous area. The light reflected from the sample was directed to a photomultiplier (PMT, H5783-01, Hamamatsu, Japan) through a bandpath filter; thus, only the probe light could enter the entrance window. The PMT output signal was transmitted to a digital oscilloscope (LeCroy 9310-L, LeCroy, Chestnut Ridge, NY, USA), and then it was processed using a computer. The signal acquisition was triggered using a photodiode hit by the pulse of the pump laser. A total of 150 and 400 traces were averaged to record fast kinetics (see Section 3.2) and slow kinetic experiments, respectively.

To control the relative humidity (RH) of the PYP biofilm, the sample, along with optical accessories and the PMT, was placed in an isolated chamber (Figure 3). The RH was adjusted with supersaturated potassium chloride solution at a fixed temperature (78%, 25 °C) [4]. To maintain the temperature and RH during the measurements, these parameters were continuously monitored with a digital hygro-thermometer (8709 Thermo-Hygrometer, Carl Roth GmbH & Co. KG, Karlsruhe, Germany).

## 3. Results and Discussion

### 3.1. Structure of the PSi Microcavity Multilayer

In Figure 4a, a cross-section of the multilayer structure is shown. The darker layers correspond to the high-porosity regions where the nanopores have thinner walls, whereas the lighter layers correspond to low porosity (thicker pore walls). The top, high-porosity layer is missing due to mechanical damage suffered during the cross-sectional cut. The lighter zone at the very top of the figure shows the second, low-porosity layer surface at a small viewing angle. In Figure 4b, the top view of several layers is seen after they became visible due to the partial peeling off of the top layer(s). It offers an excellent view of the internal porous structure of the multilayered structure, showing that the average pore diameter for the high-porosity layers is in the range of 100 nm.

### 3.2. PYP-Functionalization of PSiMcs

To characterize the optical properties of the microcavities, reflectance spectra of the untreated and PYP-infiltrated PSiMcs were recorded in the 400–800 nm range. Both reflectance curves (Figure 5) show major resonances corresponding to low-reflectance (‘negative’) peaks in the red wavelength regime (between 600 and 700 nm). Their shape and location are characteristic of the porous silicon microcavities. A spectral shift was observed between the reflectance spectra of the untreated and bio-functionalized microcavities. The peak position of the cavity mode shifted from 645 to 667 nm, corresponding to a 22 nm shift towards longer wavelengths upon functionalization, as a consequence of the increase in the refractive index of the porous silicon layers caused by the infiltration of the PYP protein.

The reflectance spectra of the untreated and PYP-functionalized samples were also simulated as follows. The effective refractive index of the low- and high-porosity layers, n_PSi_, was calculated for the non-functionalized sample by using the two-component Bruggeman effective medium approximation equation [28]:(1)$$\left(1-p\right)\cdot \frac{n_{Si}^{2}-n_{PSi}^{2}}{n_{Si}^{2}+2\cdot n_{PSi}^{2}}+p\cdot \frac{n_{A}^{2}-n_{PSi}^{2}}{n_{A}^{2}+2\cdot n_{PSi}^{2}}=0$$
where p is porosity, n_Si_ is the wavelength-dependent refractive index of bulk silicon, and n_A_ is the refractive index of air, equal to 1. For the functionalized sample, the three-component Bruggeman effective medium approximation was used [29], where the contribution of the pores was split into two terms, one corresponding to the volume fraction, β (0 ≤ β ≤ 1), occupied by the PYP protein with refractive index n_PYP_, taken as 1.53, and the other fraction, (1 − β) still occupied by air:(2)$$\left(1-p\right)\cdot \frac{n_{Si}^{2}-n_{PSi}^{2}}{n_{Si}^{2}+2\cdot n_{PSi}^{2}}+p\;\left(1-\beta \right)\cdot \frac{n_{A}^{2}-n_{PSi}^{2}}{n_{A}^{2}+2\cdot n_{PSi}^{2}}+p\;\beta \cdot \frac{n_{PYP}^{2}-n_{PSi}^{2}}{n_{PYP}^{2}+2\cdot n_{PSi}^{2}}=0\;$$

Once the refractive indices for the two layer types are calculated with the porosities as input parameters, the reflectance spectrum of the multilayer can be calculated using the transfer matrix formalism [30], taking into account the thickness of the low- and high-porosity layers as determined from the electron micrographs, such as shown in Figure 4a. Figure 5 shows the calculated spectra based on the above two models described using Equations (1) and (2). The porosities were taken as p_L_ = 0.772 and p_H_ = 0.870 for the low- and high-porosity layers, respectively. Except for minor details, the calculated spectra reproduced the measured spectra well and, in particular, the spectral shift of the resonance dip as a result of functionalization. From Equation (2), it follows that the protein layer occupies ~8% of the volumes of the pores (β = 0.079).

Based on these results, we can deduce that infiltration of the protein into the cavities took place, changing the optical properties of the porous silicon microcavities, and offering the possibility of protein-based all-optical modulation on these samples.

We can give a rough estimate of the amount of protein filling the pores and the resulting optical density (absorbance) of the protein component at the 460 nm excitation wavelength, in order to assess whether the entire depth of the infiltrated multilayer sample is photoexcited. The volume occupied by the protein in a total volume with surface area F and depth d is estimated as V = βFd(p_H_ + p_L_)/2. The number of protein molecules in this volume is then obtained as V/V_PYP_, where V_PYP_ can be estimated based on the crystal structure of PYP from the Brookhaven Protein Data Bank (file 2ZOH.pdb). Hence, the number of moles of PYP present in the volume V is V/(V_PYP_N_A_), N_A_ being Avogadro’s number, so the corresponding average molar concentration of PYP in the multilayer structure is c = V/(V_PYP_N_A_Fd) = β(p_H_ + p_L_)/(2V_PYP_N_A_). This gives with V_PYP_ = 40 nm^3^ = 4 × 10^−23^ L, β = 0.079, p_L_ = 0.772, p_H_ = 0.870 a molar concentration of PYP c = 2.7 mM. One can then calculate the absorbance of the protein sample in the total depth of the porous silicon structure as A = c ε d. Using the values d = 2.5 µm (see Figure 4) and ε (460 nm) = 2.8 × 10^4^ M^−1^cm^−1^ [17], it follows that A = 1.9 × 10^−2^, assuring that the whole depth of the sample is efficiently reached by the laser flash exciting the photocycle of PYP.

### 3.3. Demonstration of All-Optical Modulation on the Biofunctionalized PSiMc Samples

Next, we focused on the demonstration of an all-optical modulation by the PYP-infiltrated PSiMc samples. Upon light excitation of PYP at 460 nm, the photoactivation of the protein and the resulting change in the reflectance of the functionalized PSiMc was monitored by using a time-resolved experimental setup. The modulated signals were acquired at two different timescales, 0.5 and 400 ms, respectively. The signals were recorded from the same region of the sample where the reflectance spectra were measured. The wavelength of the probe light beam (675 nm) was chosen to match the steeply changing region of the spectrum beyond the main cavity peak (see Figure 5), in order to sensitively detect any light-induced shift of the resonance spectrum. The intensity changes registered upon light excitation showed complex, bipolar relaxation kinetics, with a fast negative and a slow positive phase (Figure 6). The decrease and successive increase in the intensity of the reflected light is interpreted as a consequence of a rapid red-shift of the reflection spectrum of the sample, followed by a slow blue-shift, respectively, before decaying to the initial level. This row of events is consistent with a rapid formation of the red-shifted pR, and a subsequent accumulation of the blue-shifted pB intermediates, in agreement with the known photocycle of PYP, and the Kramers–Kronig relations [13]. After the fast, unresolved drop of the signal, two exponential components could be fitted to the decay of the negative, and one for the positive phase, with a high goodness-of-fit (R^2^_fast_ = 0.83, R^2^_slow_ = 0.88). The time constants of the two negative components at the 0.5 ms timescale were τ_1_ = 24 µs and τ_2_ = 281 µs (Figure 6b,c), while τ_3_ = 131 ms for the positive phase registered on the millisecond scale (Figure 6d). The values are close to the ones obtained by evaluating absorption kinetic signals measured earlier under different conditions [13,31]. The component amplitudes were −0.012 V for the fast signal on a 4 V baseline (Figure 6a), whereas −0.002 V on a 2 V baseline for the slow signal (Figure 6d), resulting in a 0.2–0.3% relative output signal modulation, while the amplitude of noise was 0.001 V. The 0.3% signal amplitude change at 675 nm could be well reproduced by changing in our calculations the refractive index of PYP, n_PYP_ from 1.53 to 1.5315, as a result of light excitation, and recalculating the reflectance spectrum using Equation (2). With these numbers, the resulting redshift of the calculated reflectance spectrum is ~0.05 nm. The estimated refractive index change of PYP, 1.5 × 10^−3^, is in good agreement with our earlier results [13,14].

## 4. Conclusions and Outlook

The nearly 30 nm shift of the resonance dip upon protein treatment of the porous silicon structure is a clear indication of its successful functionalization by PYP. The size of the shift is also fully consistent with the results of model calculations based on the geometric parameters of the PSi structure and the estimated refractive index of the dried PYP layer. In order to demonstrate light-induced reflectance changes, the probe wavelength of 675 nm was selected by considering that here PYP shows negligible absorption, at the same time; the resonant dip in the reflectance spectrum of the PYP-functionalized PSi has a significant slope. The light-induced traces are consistent with those of absorption kinetic signals of PYP measured under other conditions and indicative of a successful light-controlled modulation of the reflectance properties of this proof-of-concept protein-doped silicon structure. For its practical application as a composite photonic device, however, the modulation depth of the reflected light has to be considerably improved. This can, in principle, be done by improving the sample-preparation procedure to increase the present 8%-efficiency of protein-doping or by sharpening the resonance dip of the microcavity to exploit more efficient all-optical modulation by the same light-induced refractive index change. According to recent results, it is possible to prepare high-Q resonant structures by further increasing the number of [HL] double layers [32], controlling the temperature of anodization [33], or combining PSi structures with layers of porous alumina [34], reaching Q-factors higher than 950 [33]. Although, filling the holes of PSi structures leads to a certain decrease in such high Q-factors [35], a significant improvement of the modulation depth is expected in the future, using one or more of the above techniques. The recently demonstrated ultrafast refractive index changes of PYP films [16] hold additional possibilities for future photonic applications of such protein-doped PSi materials.

## Figures and Tables

**Figure 1 nanomaterials-13-02070-f001:**
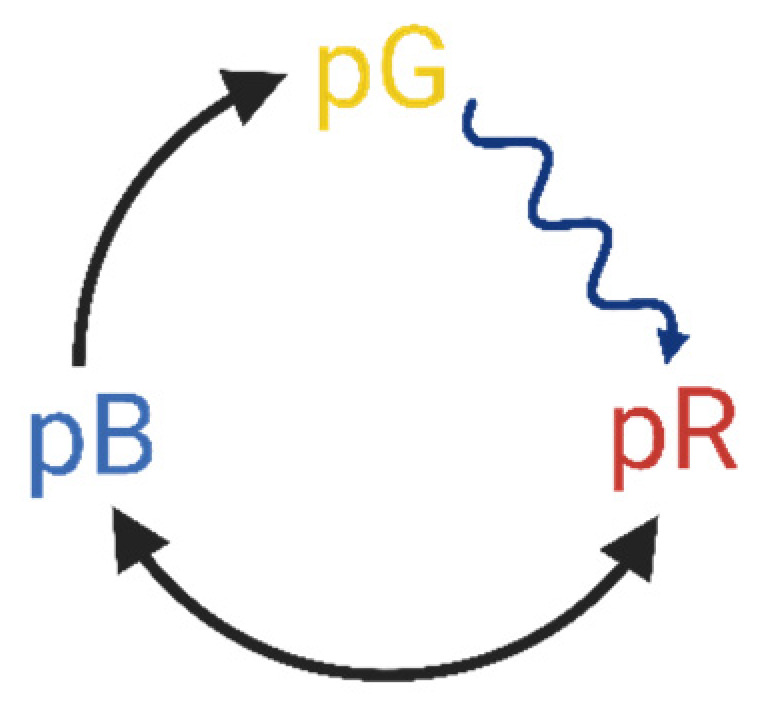
Simplified scheme of the photoactive yellow protein (PYP) photocycle with the resting state (pG) and intermediates of red- and blue-shifted (pR and pB) absorption spectra. The arrows indicate the direction of the photocycle, wavy arrow representing the light-induced reaction, while the back-and-forth arrow indicates an equilibrium reaction between pB and pR. This figure was created with BioRender.com.

**Figure 2 nanomaterials-13-02070-f002:**
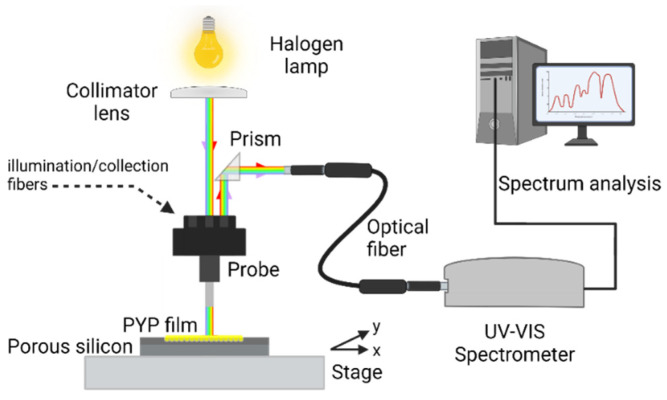
Schematic illustration of the experimental setup for the reflectance measurements of untreated and PYP-treated PSiMc samples. (Spot-size of illumination on the sample: 2 mm in diameter, distance of the lightguide bundle from sample surface: ~1 mm). This figure was created with BioRender.com.

**Figure 3 nanomaterials-13-02070-f003:**
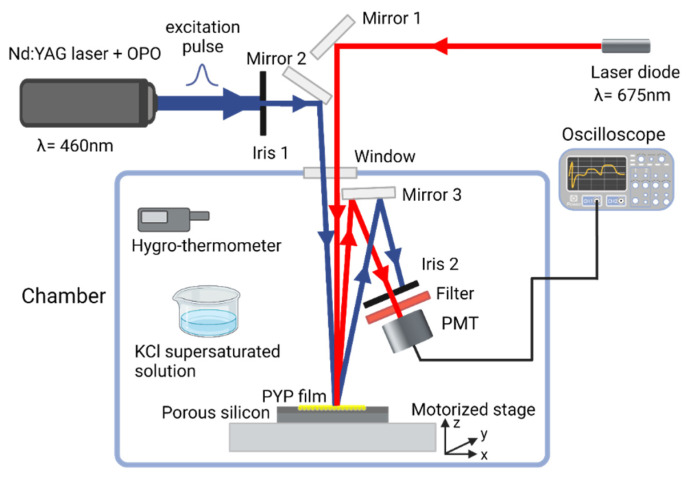
Schematic illustration of the experimental setup used for measurements of PYP-based all-optical modulation. The PYP-functionalized porous silicon sample was placed in an isolated chamber, where the relative humidity was set and monitored. The protein film was illuminated by a blue light pulse (λ = 460 nm) from a Nd:YAG laser with an OPO extension, and probed with continuous red light from a laser diode (λ = 675 nm). The reflected light was filtered and then coupled to a photomultiplier tube (PMT). The signal was transmitted to a digital oscilloscope and processed. This figure was created with BioRender.com.

**Figure 4 nanomaterials-13-02070-f004:**
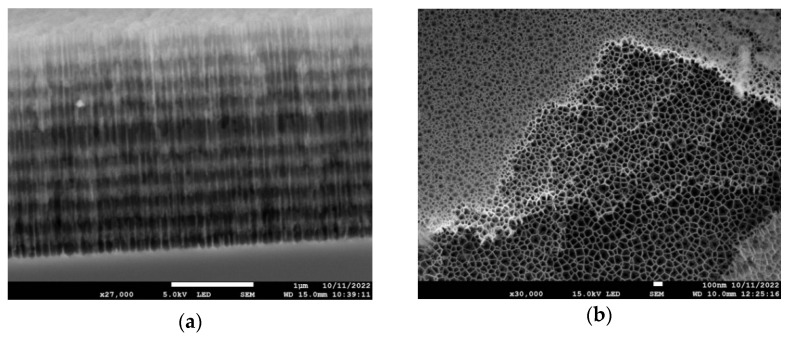
Scanning electron microscopy images of the porous silicon microcavity samples. (**a**) Cross-sectional view and (**b**) top-view with layer structures.

**Figure 5 nanomaterials-13-02070-f005:**
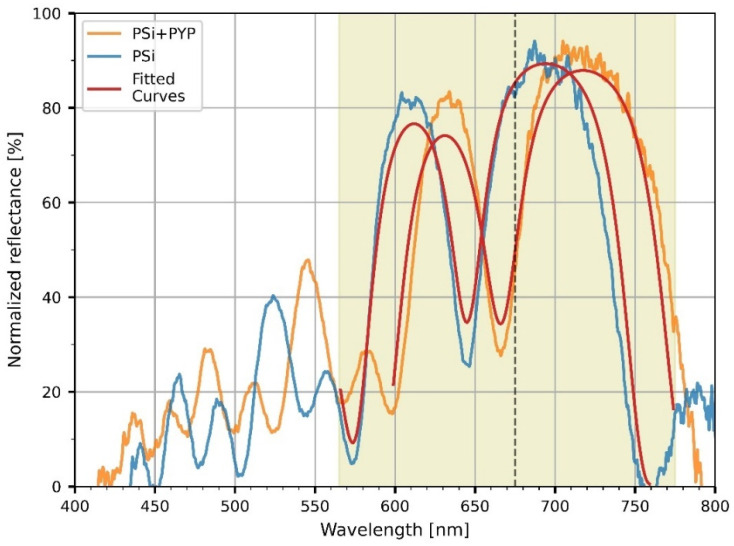
The measured reflectance spectra of porous silicon microcavity (PSiMc) before (blue) and after (orange) functionalization by PYP, and the corresponding calculated spectra based on Equations (1) and (2) (red). For details of the model calculations, see text. Thin dashed line represents the wavelength of the probe light at 675 nm for the all-optical modulation experiments.

**Figure 6 nanomaterials-13-02070-f006:**
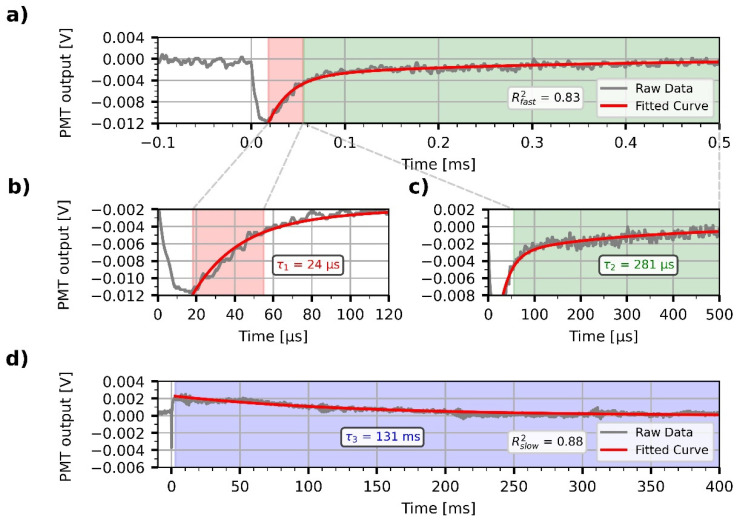
All-optical modulation by PYP-functionalized porous silicon microcavity. For both the fast, negative (**a**) and the slow, positive (**d**) phases, the modulation signals followed the reaction kinetics of the photocycle of the protein, with exponential decay time-constants (τ_1,_ τ_2,_ τ_3_) comparable with those of the PYP photocycle (**b**–**d**). Exponential components were fitted with high goodness-of-fit (R^2^_fast_ = 0.83, R^2^_slow_ = 0.88).

## Data Availability

The original data are available upon request from the corresponding authors.

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
