# Peer review of "Microsecond All-Optical Modulation by Biofunctionalized Porous Silicon Microcavity"

_nanomaterials, 2023, doi:10.3390/nano13142070_

Round 1

Reviewer 1 Report

In this manuscript, authors report the optical modulation of reflectance of porous silicon (PSi) to which water-soluble chromoprotein, photoactive yellow protein (PYP) is doped. Laser excitation of doped PYP changes the reflectance, while the magnitude is small. However, the kinetics of reflectance change well agreed with the photocycle kinetics of PYP, which strongly suggests that the modulation is derived from the properties of PYP intermediates. The observation is clear, and this reviewer recommend the publication of this manuscript after consideration of several minor points.

(1) Figure 1: Arrow for photoreaction should be different from those of thermal reactions (e.g. wavy line).

(2) Line 232: "0.004" should be "0.002", according to Figure 6d. The change of reflectance for formation of pR (0.003%) is significantly greater than that of pB (0.001%), suggesting that refractive index of pB is closer to pG than pR. However, the absorption spectrum as well as tertiary structure of pR is similar to pG, while those of pB is substantially different. Are there any explanations for this controversy?

(3) Line 263: In this context, "protein" should be specified as PYP.

(4) It is possible that the absorbance of doped PYP at 460 nm is too high for 460-nm laser to penetrate into PSi and excite all PYP molecules. Can the optical density be estimated?

Reviewer 2 Report

The paper is devoted to all-optical modulation of light by the photochromic composite microcavity. This research is up-to-date and can be useful for investigation of biological substances. Therefore the paper can be published after resolving some questions:

Line 83: please explain the role of glycerol in the electrolyte.

Line 86: "dielectroc" -> "dielectric"

Line 90: "The top layer was of high porosity, allowing easy incorporation of PYP macromolecules." PYP macromolecules are expected to fill all the structure, not the top layer only. Please comment if PYP can really easy incorporate the structure in the case of "H" top layer.

Line 104: please specify the parameters of the optical fibers: core diameters, the distance between core centers, numerical aperture. It is crucial because the spectrum of the microcavity has a strong andular dependency. The light comes from one fiber and reflecting to another, so the effective incident angle is not perpendicular.

Line 106: Why the prism is needed? You have 7 fibers: one for the illumination, six for collection. The light is already separated. Collection fibers typically are bundled into a separate port which can be connected to the spectrometer directly.

Line 116: Please specify the size of the spot where the spectrum was measured, an analogue of the beam waist.

Line 167: You have mentioned that the top layer has the high porosity. In fig. 4 both images show that the top layer has the low porosity. It is lighter in (a) and has smaller pores in (b). Please explain.

Lines 184 and 187: Authors use Bruggeman approximation (1) in the first case and the simplest approximation, so called Newton approximation (2) in the second case. In fact, the composite medium is three-component: Si, air, PYP. Please explain why you don't use three-component models. Please explain why you use different effective models in these cases.

Line 235: Please discuss the thermal mechanism of the modulation: the variation of the refractive index of pSi or PYP caused by heating with the pump laser beam, and explain why the modulation presented here is not caused by the ordinary heating.

Line 276: Q-factors higher than 950 are achievable for pure samples. The addition of substances will significantly decrease the Q-factor. This effect is small for presented here low Q. So the perspectives described in the end op the paper are too optimistic.
